# Comparative Genomics of Human- and Wastewater-Derived CPE Isolates in The Netherlands Reveals Shared and Complementary Characteristics

**DOI:** 10.3390/microorganisms14010016

**Published:** 2025-12-20

**Authors:** Hetty Blaak, Sandra Witteveen, Angela de Haan, Marga G. van Santen-Verheuvel, Merel A. Kemper, Ana Maria de Roda Husman, Antoni P. A. Hendrickx, Heike Schmitt

**Affiliations:** 1Centre for Zoonoses and Environmental Microbiology, Centre for Infectious Disease Control (CIb), National Institute for Public Health and the Environment (RIVM), P.O. Box 1, 3720 BA Bilthoven, The Netherlandsheike.schmitt@rivm.nl (H.S.); 2Centre for Infectious Diseases Research, Diagnostics and Laboratory Surveillance, Centre for Infectious Disease Control (CIb), National Institute for Public Health and the Environment (RIVM), P.O. Box 1, 3720 BA Bilthoven, The Netherlandsantoni.hendrickx@rivm.nl (A.P.A.H.)

**Keywords:** multi-drug resistance, carbapenemase-producing Enterobacterales, wastewater-based surveillance, clinical surveillance, whole-genome MLST

## Abstract

Wastewater-based surveillance is gaining interest worldwide as a complementary tool informing human surveillance of pathogens, among which are antibiotic-resistant bacteria. The current study investigated whether CPE detected within the Dutch human CPE surveillance could be identified among isolates that were independently retrieved from Dutch wastewater. Whole genomes of 203 wastewater- and 1278 human-retrieved isolates were compared using whole-genome multilocus sequence typing (wgMLST), resistome, and plasmid analyses. Overall, 25 clusters (16 *E. coli*, 9 *K. pneumoniae*) with genetically highly related variants from both niches were detected. The maximum allelic difference between human- and wastewater-derived isolates in clusters was on average 0.51% (23/4503 alleles, *E. coli*) and 0.22% (11/4978 alleles *K. pneumoniae*). For seven clusters, in-depth plasmid analysis was performed, showing highly homologous (87–100%) carbapenemase-containing plasmids from human- and wastewater-retrieved isolates. Six clusters contained wastewater and human isolates that were spatiotemporally related. The sequence identity at chromosomal and plasmid level confirms the presence of human-associated CPE variants in wastewater. Ongoing comparisons between isolates from the national human CPE surveillance and wastewater surveillance will shed more light on the added value of wastewater-based surveillance for monitoring of CPE and other (emerging) antibiotic resistances.

## 1. Introduction

Infections with carbapenemase-producing Enterobacterales (CPE) are considered a major threat to public health globally [1]. Although endemic in humans in some parts of the world, in Northern European countries, the incidence of CPE is still relatively low [2]. In the Netherlands, carriage of CPE is associated with travel abroad and, in particular, a hospital stay abroad [3]. Surveillance of multi-drug-resistant microorganisms (MDRO) at a national scale is important to aid policymakers in making decisions to limit the spread of antimicrobial resistance. In the Netherlands, a human CPE surveillance program has been in place since 2012 [4,5,6]. CPE identified in this surveillance mostly originate from persons with infections or asymptomatic carriers in high-risk groups. Therefore, the frequency of asymptomatic carriage among the general population based on this surveillance remains largely unknown.

For the past couple of years, surveillance through measurements of wastewater has gained increasing interest globally as a tool to monitor the prevalence of (fecal) pathogens, among which MDROs, in the general population [7,8,9,10,11,12,13,14,15]. Global applications of wastewater-based surveillance vary from inventorying entire populations of microorganisms (microbiomes) and their resistance (resistomes) through metagenomic approaches [7,8,9], to the determination of the prevalence of specific resistance genes [10] or specific bacterial species and resistance combinations using culture techniques [11,12,13,14,15]. In the recent recast of the European urban wastewater treatment directive (UWWTD), monitoring of antimicrobial resistance has been included within urban wastewater surveillance [16]. The main advantage of wastewater-based surveillance is that it offers the possibility to screen large groups of people at the same time with a high sensitivity, thereby diminishing the need for screening individuals. Wastewater surveillance is therefore particularly useful for the monitoring of rare, partially asymptomatic (i.e., opportunistic) pathogens. The most practical location to monitor wastewater, and in high-income countries, the most used location, is at wastewater treatment plants (WWTPs), where sewage is collected and treated before it is discharged into surface water [17]. The WWTP catchment (i.e., the part of the sewer system connected to a WWTP) represents the whole population connected, if automated sampling is used to obtain 24 h volume-proportional composite samples.

Previously, we have performed several studies to detect, quantify, and characterize CPE in Dutch municipal wastewater and hospital wastewater. These studies have confirmed that wastewater is a sensitive matrix for the detection of these rare bacteria and indicated the value of wastewater surveillance [11,18]. The goal of the current study was to further validate the use of wastewater-based surveillance by investigating the degree of overlap between CPE genotypes detected in wastewater and in humans. In this first exploratory study, available sequence data of CPE variants from previous wastewater studies were compared with those obtained in the Dutch national human CPE surveillance. Comparative analyses entailed whole-genome multilocus sequence typing (wgMLST) and in-depth plasmid analysis. Insights into the relation and the level of similarity of genotypes between the two compartments are crucial to fully understand the merits of wastewater surveillance and to investigate its added value in determining the prevalence and spread of CPE in the general population.

## 2. Materials and Methods

### 2.1. Origin of Human-Retrieved CPE Isolates

For the Dutch National CPE Surveillance program, medical microbiology laboratories routinely send Enterobacterales isolates with genotypical or phenotypical evidence of carbapenemase production (since July 2016) or a meropenem minimum inhibitory concentration (MIC) of >0.25 mg/L and/or an imipenem MIC of >1 mg/L (before July 2016) to the National Institute of Public Health and the Environment (RIVM). The MIC threshold for submission is based on the national guideline for the detection of carbapenemase production. It is chosen so that all variants with acquired resistance are included (as opposed to only variants with clinical resistance), as these represent a reservoir for the spread of antibiotic resistance genes [4,5]. The human CPE cases included in the surveillance consist of persons having an infection with CPE, as well as asymptomatic carriers who were screened at a healthcare institution for other reasons. The most frequent reason for screening was having a recent history of hospitalization abroad. In the majority of cases, patient cultures had been taken at hospitals (68%), followed by general care practices (15%), and nursing homes (2.5%). For 13% of the cases, data on the site where the patient cultures were taken was missing. Overall, 1278 CPE originating from the surveillance program were included in the present study: 551 *Escherichia coli* and 727 *Klebsiella pneumoniae* complex (*K. pneumoniae* and *Klebsiella variicola*) isolates, which were collected between 29 February 2012 and 31 December 2020 (Appendix A). Human-derived isolates were included from the entire time range because the study was a first exploratory investigation of the correlation between human- and wastewater-derived isolates, and aimed not to a priori rule out the possibility of uncovering potential similarities between isolates obtained in different years. Only the first submitted *E. coli* or *K. pneumoniae* complex isolates with a given carbapenemase allele were included per person per year.

### 2.2. Origin of Wastewater CPE Isolates

CPEs were isolated from wastewater samples as part of different study projects carried out between 2015 and 2018 [11,18]. One of these studies involved a sampling campaign at 100 WWTPs scattered across The Netherlands, of varying sizes and differing in terms of the presence/absence of healthcare institutions [11]. The selected WWTP represented 29% of all Dutch WWTP operational at that time (*n* = 341) and together served approximately 40% of the Dutch population. Another study consisted of a sampling campaign at eight hospitals, one nursing home, and the WWTPs receiving the wastewater from these institutions [18].

In short, from each sample multiple volumes (for the purpose of obtaining quantifiable colony counts) were filtered through 0.45 µm pore-size membrane filters in two-fold, of which half was placed on ChromID CARBA agar and the other half on ChromID OXA-48 agar (BioMérieux, Zaltbommel, The Netherlands). All cultures were incubated for 4 to 5 h at 37 °C, followed by 18 to 24 h at 44 °C. The selective temperature of 44 °C was used to inhibit the growth of background biota. Presumptive CPE colonies were streaked on the type of medium they were isolated on to confirm their capacity to grow on the selective media. Thus confirmed carbapenem-resistant isolates were identified to the species or genus level using MALDI-TOF MS. All Enterobacterales were screened for the presence of *bla*_NDM_, *bla*_KPC_, *bla*_OXA-48_-like, and carbapenemase-type *bla*_GES_ genes using PCR and Sanger sequencing. From carbapenemase-positive *E. coli* and *K. pneumoniae* complex isolates, a selection was characterized using whole-genome sequencing (WGS) within the context of the original studies. Selection criteria were diverse and dependent on the research questions of the individual studies, largely aiming to maximize diversity among the selection (to study variability of CPE from 100 WWTPs) or to minimize diversity (to confirm homogeneity between isolates from different samples in the study on institutional wastewater). Selections focused on *bla*_NDM_*-* and *bla*_KPC_-positive isolates because of the clinical relevance of this type of resistance. Additionally, a small proportion of isolates (*n* = 25) was sequenced specifically for the current comparison with human isolates. These were randomly picked among *bla*_NDM_*-* and *bla*_KPC_-positive isolates from samples that were not among those sequenced previously. Overall, whole-genome sequences were available for 307 isolates, which were mostly isolated from municipal wastewater sampled at WWTPs (*n* = 217 isolates from 81 samples from 37 WWTPs) and hospital wastewater (*n* = 89 isolates from 19 samples from 6 hospitals) (Appendix A). For comparison with the human isolates, only ‘unique’ isolates were included in the final analyses, i.e., only one randomly selected wgMLST variant per sample location per year, and if locations had been sampled at multiple timepoints, at least 4 months apart. This resulted in 203 included wastewater isolates: 152 *E. coli* and 51 *K. pneumoniae* (from 95 samples at 44 locations; Appendix A).

### 2.3. Illumina and Nanopore Whole-Genome Sequencing

Human and wastewater *E. coli* and *K. pneumoniae* isolates were subjected to WGS using the Illumina HiSeq 2500 (BaseClear, Leiden, The Netherlands). Nanopore sequencing was performed using an in-house developed protocol as described previously [19]. In brief, the Oxford Nanopore protocol SQK-LSK108 (https://community.nanoporetech.com, accessed on 12 March 2024) and the expansion kit for native barcoding EXP-NBD104 were used (Oxford Nanopore Technologies, Oxford, UK) for isolates from 2012 to 2018. To obtain larger DNA fragments for isolates, from 2019 onwards, the protocol SQK-LSK109 was followed (Oxford Nanopore Technologies, Oxford, UK). The library was loaded onto a MinION flow cell (MIN-106 R9.4.1), and the sequence run was started on a MinION device (Oxford Nanopore Technologies, Oxford, UK, accessed on 12 March 2024) without live-base calling enabled. After the sequence was run, basecalling and de-multiplexing were performed using Albacore v2.3.1, and a FASTA file per isolate was extracted using Poretools v0.5.1. Illumina (>25x coverage) and Nanopore (>20x coverage) data were used in a hybrid assembly performed by Unicycler v0.4.4. The resulting contig files were annotated using Prokka v1.14.6. The antibiotic-resistance gene profile and plasmid replicon compositions in all of the isolates were determined by interrogating the ResFinder (v3.1.0) and PlasmidFinder (v2.0.2) databases available from the Center for Genomic Epidemiology. For ResFinder, a 90% identity threshold and a minimum length of 60% were used as criteria, whereas for PlasmidFinder, an identity of 95% was utilized. The resulting Illumina- and Nanopore-derived data, such as resistance genes, replicons, MLST and wgMLST profiles, and circular plasmids, were imported into BioNumerics version 8.1 (Applied Maths, Sint-Martens-Latem, Belgium) for subsequent comparative genomic analyses (see below).

### 2.4. Minimum Spanning Tree, MLST, and wgMLST Analyses

The BioNumerics software was used to generate a minimum spanning tree (MST) from wgMLST profiles. The categorical coefficient was used to calculate the MST, which was based on in-house *E. coli* and *K. pneumoniae* wgMLST schemes [19] made in the SeqSphere software version 6.0.2 (Ridom GmbH, Münster, Germany). The in-house *K. pneumoniae* wgMLST scheme comprises 4978 genes (3471 core-genome and 1507 accessory-genome targets) using *K. pneumoniae* MGH 78,578 (NC_009648.1) as a reference genome. The in-house *E. coli* wgMLST scheme comprises 4503 genes (3199 core-genome and 1304 accessory-genome targets) using *E. coli* 536 (CP000247.1) as a reference genome. Genetic clusters were defined as 2 or more isolates varying with ≤25 wgMLST allele differences for *E. coli* and ≤20 wgMLST allele differences for *K. pneumoniae* complex as described previously [16]. Clusters containing both human and wastewater isolates were analyzed in more detail regarding metadata (location and date of origin), resistance genes, replicons, and plasmids.

### 2.5. Analysis of the Geographic Relationship Between Human and Wastewater Isolates

For isolates in genetic clusters, the existence of a geographical relation between the human and the wastewater isolates was analyzed retrospectively. An exact geographical match between isolates was defined as the situation where the residential area of the carriers and/or the healthcare institution where the human isolate was isolated, was located within the catchment area of the WWTP where the wastewater isolate was isolated. This was performed by coupling zip-code data of CPE carriers available in the human surveillance database with publicly available data on zip codes within sewer catchments provided by Statistics Netherlands [20]. Used were people’s partial zip codes (partial, because of the General Data Protection Regulation) and the complete zip codes of the hospital (or medical microbiology laboratory if the hospital was not registered), wherever this information was available from the surveillance database. The partial zip codes are the first four digits of zip codes (which consist of four digits and two letters). The partial zip codes correspond to neighborhoods, and for the majority (80%), fall completely within a single sewer catchment. If no exact geographical match was observed, the distance between the residential area and the healthcare institution of the human case and the WWTP was estimated. When the distance was within 25 km (as the crow flies), the isolates were considered to be a “near geographical match”.

### 2.6. Plasmid Reconstruction by Read Mapping

The isolates from three *E. coli* wgMLST clusters (designated Eco4, Eco11, Eco16) and four *K. pneumoniae* wgMLST clusters (Kpn2, Kpn3, Kpn4, Kpn6) were selected for plasmid analyses. Selection was performed to include: (1) small as well as large clusters, (2) clusters varying with respect to the allelic distance between human and wastewater isolates (see Table 1), (3) clusters containing human and wastewater isolates that were retrospectively geographically matched, and (4) isolates originating from very different places in the country. From these seven clusters, the whole genome of one randomly selected human isolate was obtained using Nanopore long-read sequencing to obtain complete plasmid sequences. Next, CLC Genomics Workbench version 12.0 software (www.qiagenbioinformatics.com, accessed on 12 March 2024) was used to reconstruct plasmids in isolates that were not sequenced with Nanopore long-read sequencing. For this, the complete plasmids obtained by long-read sequencing were used as a scaffold to map the trimmed short-reads of isolates that were from the same genetic wgMLST cluster. A plasmid was scored as “present” in an isolate when reads mapped to the reference plasmid of interest and ≥95% of the consensus sequence size in kilobases was reconstructed, as described previously [21]. Linear DNA fragments of <2.5 kb were omitted in this study.

### 2.7. Ethics Statement

The human-retrieved bacterial isolates belong to the medical microbiological laboratories participating in the Dutch National CPE Surveillance program and were obtained as part of routine clinical care in the past years. Since no identifiable personal data were collected and data were analyzed and processed anonymously, written or verbal patient consent was not required. According to the Dutch Medical Research Involving Human Subjects Act (WMO), this study was exempt from review by an Institutional Review Board.

### 2.8. Data Availability

The whole genome sequence data generated in this study are available in the Sequence Read Archive (SRA) under projects PRJEB35685, PRJEB42331, PRJEB73354, PRJNA634885, PRJNA804679, PRJNA980147, PRJNA1101444, PRJNA1143178, and PRJNA1247921. Accession numbers are provided through Appendix A.

## 3. Results

### 3.1. Distribution of Genetic Clusters Containing Human and Wastewater E. coli and K. pneumoniae Isolates

For both *E. coli* and *K. pneumoniae*, human and wastewater isolates were located interspersed in the wgMLST minimum spanning trees (MST, Figure 1 and Figure 2). Multiple distinct genetic clusters were observed, containing human as well as wastewater isolates: 16 clusters for *E. coli* and 9 clusters for *K. pneumoniae*. Overall, nearly a quarter of the wastewater isolates clustered with human isolates (49/203, 24%), and 10% of the human isolates clustered with wastewater isolates (124/1278). The allelic distance within the *E. coli* clusters was larger than within the *K. pneumoniae* clusters (on average 19 of 4503 alleles, 0.42% vs. on average 7 of 4978 alleles, 0.14%) (Table 1). The minimum and maximum allelic difference between human and wastewater isolates within clusters was, on average, 0.32% and 0.51%, respectively, for *E. coli* and 0.085% and 0.22% for *K. pneumoniae*. Genetic clusters of human and wastewater CPE comprised a variety of *E. coli* and *K. pneumoniae* MLST sequence types (ST) harboring OXA-, NDM-, and KPC-like carbapenemases, including globally epidemic *E. coli* ST38, ST167, and ST410, and *K. pneumoniae* ST11, ST307, and ST512 (Table 1, Appendix A). Despite occasional differences between isolates within clusters, resistome and plasmid replicon analysis additionally showed genetic homogeneity in the accessory genome (Appendix A).

### 3.2. In-Depth Characterization of Genetic Clusters with Human and Wastewater CPE

For seven of the genetic clusters with human and wastewater isolates, long-read sequences were obtained from one of the human CPE isolates to serve as a reference for plasmid mapping of the short-reads of the other isolates from the cluster (Figure 3). This showed that high genetic relatedness of wastewater- and human-retrieved CPE isolates was not only observed for chromosomal genes by wgMLST, but also in the reconstructed plasmids. In six of the clusters (two of three of the *E. coli* clusters and all four *K. pneumoniae* clusters), at least one of the wastewater isolates harbored the carbapenemase gene on a plasmid highly similar (87–100% homology) to that of the human reference strain (Figure 3, Table 2 and Table 3). In two of the three cases in which the reference isolate also had a *bla*_CTX-M-15_-containing plasmid in addition to the carbapenemase gene-containing plasmid, a 100% plasmid sequence match was observed for this plasmid for at least one of the wastewater isolates of the same cluster. In general, 100% identity among plasmid sequences was more often observed among *K. pneumoniae* isolates than among *E. coli* isolates.

### 3.3. Spatiotemporal Relationship Between Human and Wastewater CPE

CPE isolates with high genetic identity were observed at different moments over time, from either humans or wastewater, and at different locations (Figure 4). In 12 of the 25 wgMLST clusters with CPE from humans and wastewater, a near or exact geographical match between one or more of the human and wastewater isolates was observed: in 7/9 (78%) and 5/16 (31%) for *K. pneumoniae* and *E. coli*, respectively. When considering only the period starting from when the wastewater studies were performed (i.e., as of October 2015), the first detection of a variant occurred in wastewater for 14 (56%) clusters and in the human surveillance for 9 (36%) clusters (Figure 4). In the two remaining clusters (Eco11, Kpn5), the variant was detected only in wastewater after this time point and not seen again in humans. For 18 (72%) of the clusters, the highly identical variants were detected in humans and wastewater within the same year, for 12 (48%) clusters, this was within six months. Six wgMLST variants (represented by cluster Eco11, Eco12, Kpn3, Kpn4, Kpn6, and Kpn8) were detected in humans and in wastewater within the same year and region (Figure 4). For three of these clusters (Eco11, Kpn4, Kpn6), plasmid analyses were performed, demonstrating 50% identity of the carbapenemase-carrying plasmid for the *E. coli* isolates from cluster Eco11, and 97% and 100% identity for the two *K. pneumoniae* clusters, respectively. On the other hand, in-depth analysis of the larger clusters Eco4, Eco16, and Kpn6 also demonstrated that isolates with highly identical wgMLST profiles and conserved carbapenemase-containing plasmids can be observed in different human carriers living in different regions of the country and at timepoints that were multiple years apart.

## 4. Discussion

By comparing wastewater-retrieved isolates with isolates from the human CPE surveillance, we were able to obtain CPE isolates from humans and wastewater that were genetically highly identical with respect to chromosomal genes in wgMLST analysis that harbored near-identical plasmids. This was observed even though the human surveillance and wastewater studies were carried out independently. The sequence identity at wgMLST as well as the plasmid level suggests that the isolates in question represent the same clone, which has been detected—sometimes at multiple occasions and in different regions—in humans and wastewater. Sequence identity between wastewater and human CPE was observed for *E. coli* and *K. pneumoniae*, although a higher number of clusters was observed for *E. coli*. This is most likely explained by the higher number of *E. coli* wastewater isolates that were available for analysis.

Among the CPE from wastewater and humans were globally disseminated or epidemic MLST lineages with carbapenemase genes, such as ST11/*bla*_KPC-2_, ST307/*bla*_KPC-3,_ and ST512/*bla*_KPC-3_ *K. pneumoniae* and ST38/*bla*_OXA-48_, ST410/*bla*_NDM-5_ (and/or *bla*_OXA-181_), and ST167/*bla*_NDM-5_ *E. coli* [22,23,24,25,26,27]. The wastewater variants were interspersed between human isolates and vice versa, indicating that there is no segregation between human and wastewater strains. In particular, for *E. coli*, however, some branches of the MST contained more wastewater isolates, which is caused by the relatedness of some of the wastewater samples (i.e., samples from multiple timepoints from the same location and samples from connected hospitals and sewers at the same timepoint).

Almost a quarter of wastewater isolates clustered with human isolates, despite the fact that the wastewater had been sampled at sites and timepoints independent of the detection of human cases in the human CPE surveillance. Only for 12% of the human cases, wastewater was sampled at the relevant WWTP or hospital within 6 months. Because of this, the chance of detecting a CPE strain from identified carriers in wastewater upon excretion, i.e., with a direct causal relation, was low within the current study setup. The confirmation of human-associated CPE variants in wastewater confirms the idea that wastewater reflects carriage of antibiotic-resistant (ABR) bacteria in the human population [8,9,10,11,12,13,14,15].

In nearly half of the wgMLST clusters with human and wastewater isolates, a near geographical match was observed between the location where one (or more) of the human cases was detected and where one (or more) of the wastewater isolates was obtained. This could indicate causal relations between carriers and isolates in wastewater. However, the general lack of a temporal association (i.e., the time between detection in the person and the sampling of the wastewater was generally more than 6 months apart) makes it more likely that the detection of the variant in wastewater indicates circulation of this variant among humans within the same area. Most isolates in genetic clusters did not have geographical connections. This was the case for human and wastewater isolates within clusters, but also among human isolates or among wastewater isolates within the same cluster. This indicates that the variants or clones in question are broadly disseminated throughout the country. Geographical matches between human and wastewater isolates were more often observed within *K. pneumoniae* clusters than within *E. coli* clusters, which might be associated with a broader dissemination among the general population of CP *E. coli* compared to CP *K. pneumoniae* [11].

In more than half of the clusters, the CPE variant was detected in wastewater before it was detected in human CPE surveillance. This is remarkable, as the data set rather favored a higher chance of detecting a variant first (or only) in humans, as (a) the human screening had started earlier, (b) the number of human isolates was over six times higher than that of the wastewater isolates, and (c) the wastewater isolates were obtained at limited locations (one third of all Dutch WWTPs and eight hospitals) and few timepoints (most WWTPs were sampled only once). The early detection in wastewater is possibly related to the fact that detection in humans only occurs when the carrier comes into contact with healthcare and is tested, either because of an infection caused by the CPE or because of precaution measures taken when patients are considered to have a high risk of carriage. In other words, specific variants may circulate in the human population (and therefore in wastewater) before they are detected in human surveillance. This may also explain the detection of wastewater variants that were not detected in humans at all within the study period. Some of these variants might still be detected in humans at a later stage, but it is also plausible that some variants may never be detected in humans when they are not associated with disease and would only be detected by chance during screening. The observed discrepancies between CPE variants detected in humans and wastewater indicate that wastewater surveillance is of added value to human CPE surveillance, as it generates a broader view of the spread of and variation among CPE in the general population.

Besides the presence of CPE in wastewater from carriers that are undetected by human surveillance, discrepancies between variants in wastewater and humans may also have other causes. The exact fate of ABR fecal bacteria in the sewer, i.e., between excretion by humans and entering WWTP, is not yet known. The travel time of sewage from households and hospitals to WWTP is generally one to two days in the Netherlands, varying between WWTPs (e.g., related to distance), the sewer type (e.g., mixed or separated), rainfall, and other sewer characteristics such as the mode of transport (e.g., gravity flow or pump-based flow). The majority of the bacteria will likely survive this relatively short period in the aquatic environment and will end up alive at the WWTP. This is also confirmed by our previous studies in hospital wastewater and city sewers and their connected WWTPs [18]. Also, ABR bacteria may remain persistent in the sewer system, including plumbing and siphons, e.g., in biofilms, and additionally, new ABR variants might arise in such persistent communities through horizontal gene transfer (HGT) [28,29,30,31]. The frequency with which persistence and HGT occur in sewers and plumbing is not known. Although presumably rare, these phenomena could diminish the representativeness of ABR bacteria in WWTP influents for carriage in humans. Additionally, while municipal wastewater largely consists of household and institutional wastewater, sewers may be contaminated with bacteria from animal feces, e.g., from rats, dogs, or wild birds [32,33,34,35]. Future studies should therefore be aimed at shedding more light on the origin and fate of ABR bacteria in sewers.

The current study was a first exploratory study, which made use of independent, pre-existing sequence databases that were not designed to provide an optimal match in location and time between human and wastewater samples. The available data from wastewater were limited, as they consisted partially of the first measurements in a wastewater surveillance in development (one measurement at 100 WWTPs) and partially from a research study focused on the relation between CPE in hospitals and connected WWTPs (three to four measurements in eight hospitals and their connected WWTPs). Also, the sequence data available from these samples were limited, and isolates had been selected within the context of the previous studies. The data are not suitable therefore, to determine the sensitivity with which CPE from known human carriers can be detected at WWTPs. Neither do the current data yet explain the origin and relevance of genetic variants that were detected in wastewater but not in humans: how accurately do they reflect the synchronous situation in the general population, and will some of these variants become associated with disease at a later stage? Since 2018, a more structured and elaborate sampling scheme at WWTPs has been installed in the Netherlands [36]. Ongoing comparisons between isolates from the national human CPE surveillance and the recently established wastewater-based surveillance will shed more light on the added value of wastewater-based surveillance for monitoring of CPE and other types of (emerging) antibiotic resistance. Data from ABR wastewater surveillance will become available from more countries in the near future, as ABR monitoring in wastewater will become mandatory according to the recast of the European Urban Wastewater Treatment Directive [16].

## Figures and Tables

**Figure 1 microorganisms-14-00016-f001:**
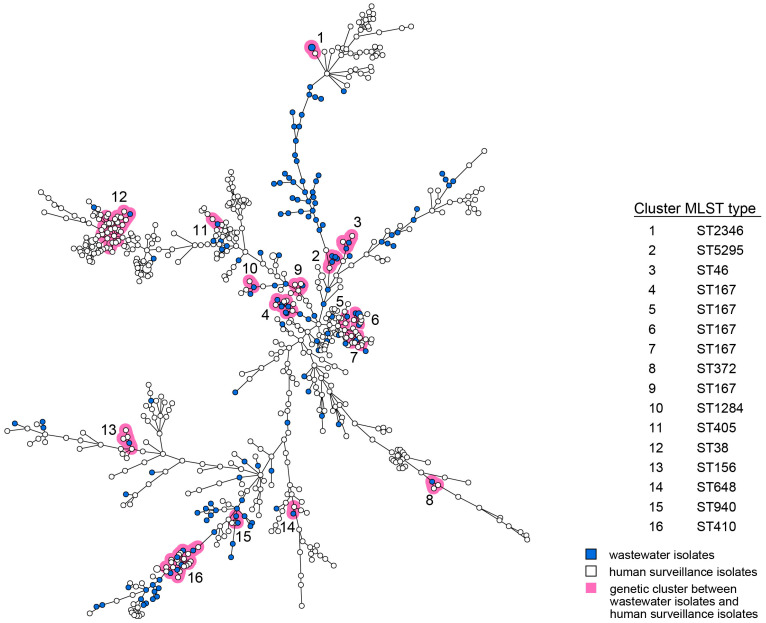
wgMLST MST of *E. coli* isolates obtained from humans and wastewater. Wastewater isolates are represented by blue nodes; human isolates by white nodes. Genetic clusters (≤25 wgMLST allele differences) are marked by pink shading and are numbered. The size of the nodes represents the number of isolates per node (generally one isolate). Lines between the nodes indicate genetic distance.

**Figure 2 microorganisms-14-00016-f002:**
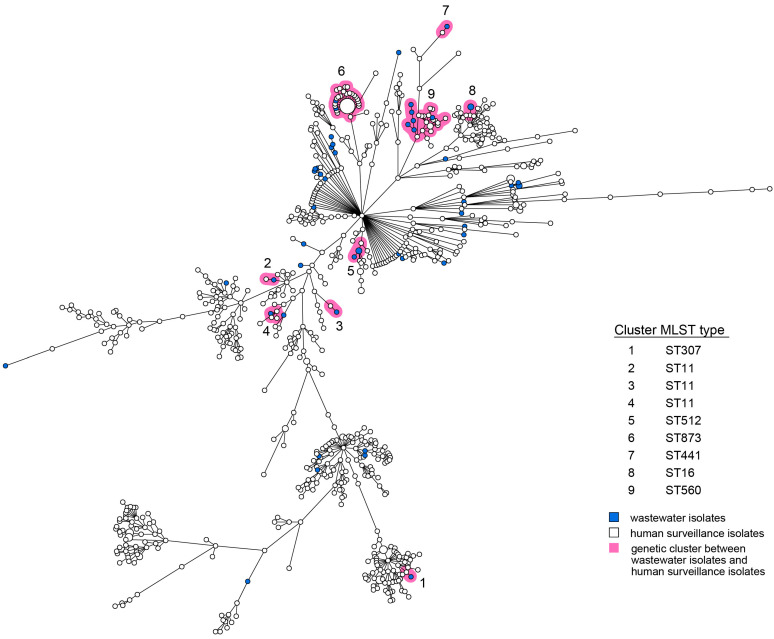
wgMLST MST of *K. pneumoniae* isolates obtained from humans and wastewater. Wastewater isolates are represented by blue nodes; human isolates by white nodes. Genetic clusters (≤20 wgMLST alleles) are marked by pink shading and are numbered. The size of the nodes represents the number of isolates per node (generally one isolate). Lines between the nodes indicate genetic distance.

**Figure 3 microorganisms-14-00016-f003:**
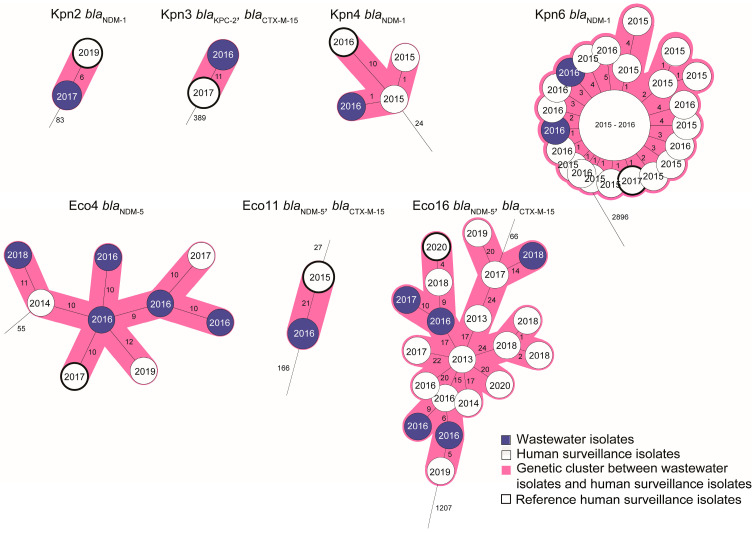
Enlarged view of wgMLST clusters selected for plasmid reconstruction. The clusters are derived from the trees displayed in Figure 1 and Figure 2. White nodes indicate human isolates; blue nodes indicate wastewater isolates. Reference isolates from which long-read sequences were obtained (one per cluster) are indicated with a thick outline. Plasmids of the other isolates in the clusters were reconstructed by mapping short reads against the long read contigs of this reference isolate. Also indicated are the years of isolation (in the nodes) and the CPE and extended-spectrum beta-lactamase (ESBL) genes carried by the isolates in the cluster. More information on the resistance gene content of the plasmids is found in Table 2 and Table 3.

**Figure 4 microorganisms-14-00016-f004:**
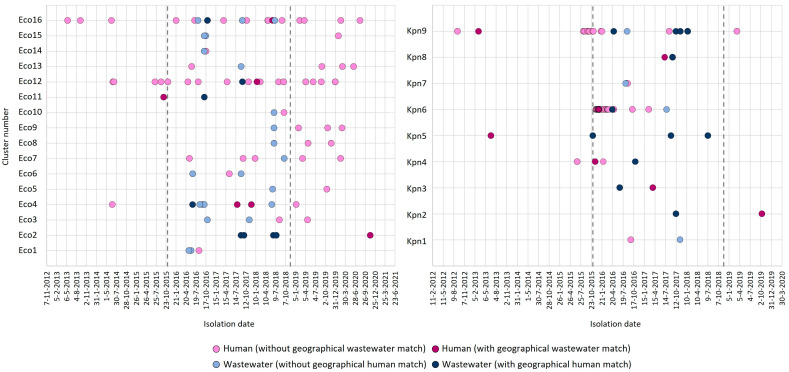
Spatiotemporal relationship between human and wastewater isolates within clusters. Shown are the timepoints at which human (blue) and wastewater (pink) isolates in genetic clusters were isolated. Dark-colored symbols indicate human and wastewater isolates isolated from the same geographical region (exact or near matches). Vertical lines show the boundaries of the time period in which the wastewater samples were obtained.

**Table 1 microorganisms-14-00016-t001:** Characteristics of genetic clusters of human and wastewater-retrieved CPE isolates.

Cluster ID	CP Allele	MLST ST	Number of Isolates		Type of WW (Number of Different Locations) ^d^	Year of Isolation		Allelic Distance ^e^		
			Human	Waste-Water		Human	Waste-Water	Average	Minimum Between Human and Wastewater	Maximum Between Human and Wastewater
Eco1	*bla* _OXA-181_	2346	1	2	WWTP (2)	2016	2016	2	3 (0.07)	3 (0.07)
Eco2	*bla* _KPC-3_	5295	1	4	WWTP (1), hospital (1)	2020	2017, 2018	6	7 (0.16)	9 (0.20)
Eco3	*bla* _NDM-5_	46	2	2	WWTP (1)	2018, 2019	2016, 2017	20	18 (0.40)	22 (0.49)
Eco4	*bla* _NDM-5_	167	4	5	WWTP (5)	2014, 2017, 2019	2016, 2018	12	10 (0.22)	16 (0.36)
Eco5	*bla* _NDM-5_	167	1	1	WWTP	2019	2020	24	24 (0.53)	24 (0.53)
Eco6	*bla* _NDM-5_	167	1	2	WWTP (2)	2017	2016, 2017	20	18 (0.40)	19 (0.42)
Eco7	*bla*_NDM-5_ or *bla*_OXA244_ ^a^	167	5	1	WWTP	2016–2020	2018	22	8 (0.18)	25 (0.56)
Eco8	*bla* _OXA-181_	372	2	1	WWTP	2019	2018	24	18 (0.40)	28 (0.62)
Eco9	*bla* _NDM-5_	167	3	1	WWTP	2019, 2020	2018	23	15 (0.33)	27 (0.60)
Eco10	*bla*_NDM-5_ or *bla*_OXA181_ ^b^	1248	1	1	WWTP	2018	2018	20	20 (0.44)	20 (0.44)
Eco11	*bla* _NDM-5_	405	1	1	WWTP	2015	2016	21	21 (0.47)	21 (0.47)
Eco12	*bla* _OXA-48_	38	20	1	hospital	2014–2019	2018	33	19 (0.42)	52 (1.15)
Eco13	*bla* _NDM-5_	156	4	1	WWTP	2016, 2019, 2020	2017	15	11 (0.24)	16 (0.36)
Eco14	*bla* _NDM-7_	648	1	1	WWTP	2016	2016	23	23 (0.51)	23 (0.51)
Eco15	*bla* _OXA-181_	940	1	2	WWTP (2)	2020	2016	15	14 (0.31)	18 (0.40)
Eco16	*bla*_NDM-5_ or *bla*_OXA181_ ^c^	410	15	5	WWTP (5)	2013–2020	2016–2018	29	5 (0.11)	46 (1.02)
Kpn1	*bla* _KPC-3_	307	1	1	hospital (1)	2016	2017	1	1 (0.02)	1 (0.02)
Kpn2	*bla* _NDM-1_	11	1	1	WWTP	2019	2017	6	6 (0.12)	6 (0.12)
Kpn3	*bla* _KPC-2_	11	1	1	WWTP	2017	2016	11	11 (0.22)	11 (0.22)
Kpn4	*bla* _NDM-1_	11	3	1	WWTP	2015, 2016	2016	6	1 (0.02)	11 (0.22)
Kpn5	*bla* _KPC-3_	512	1	3	WWTP (1), hospital (1)	2013	2015, 2017, 2018	2	2 (0.04)	4 (0.08)
Kpn6	*bla* _NDM-1_	873	36	2	WWTP (2)	2015–2017	2016, 2017	3	1 (0.02)	9 (0.18)
Kpn7	*bla* _OXA-181_	441	1	1	WWTP	2016	2016	6	6 (0.12)	6 (0.12)
Kpn8	*bla* _NDM-5_	16	1	2	WWTP, hospital	2017	2017	3	4 (0.08)	4 (0.08)
Kpn9	*bla* _KPC-2_	560	16	6	WWTP (1), hospital (1)	2012–2019	2016–2018	25	6 (0.12)	47 (0.94)

^a^ Four human and one wastewater isolate with *bla*_NDM-5_, one human isolate with *bla*_OXA-244_; ^b^ one human isolate with *bla*_NDM-5_ and one wastewater isolate with *bla*_OXA-181_; ^c^ four human and three wastewater isolates with *bla*_NDM-5_ and 11 human and two wastewater isolates with *bla*_OXA-181_; ^d^ wastewater derived from WWTP (influents) or raw hospital wastewater, in between brackets is the number of different WWTP or different hospitals, if applicable; ^e^ the average number of different loci based on all isolates in the cluster and the minimum and maximum number of different loci between a human and a wastewater isolate in the cluster, in between brackets expressed as percentage of the total number of alleles included in the wgMLST: 4503 and 4978 for *E. coli* and *K. pneumoniae*, respectively.

**Table 2 microorganisms-14-00016-t002:** Detection of plasmids in human-retrieved and wastewater-retrieved CP *E. coli* clusters.

Cluster	Origin Isolate ^a^	Isolation Date ^b^	Long Read (LR) Contig Characteristics			Results Short Read (SR) Mapping (% Match with)	
			Plasmid (Size kb)	Plasmid Replicon(s)	Carbapenemase and ESBL Genes	Plasmid 1 (Carbapenemase)	Plasmid 2 (CTX-M)
Eco4	H	27 July 2017 *	1 (46.2)	IncX3	*bla* _NDM-5_	100%	n.a.
			2 (95.2)	IncI1	*bla* _CTX-M-15_	n.a.	100%
	H	24 June 2014				0%	4.5 kb
	H	3 December 2017 *				100%	100%
	H	11 January 2019				100%	many gaps ^c^
	W	19 September 2016				100%	not present
	W	5 June 2018				100%	3.2 kb
	W	20 June 2016 *				100%	many gaps ^c^
	W	3 October 2016				100%	many gaps ^c^
	W	23 August 2016				100%	not present
Eco11	H	30 September 2015 °	1 (214)	IncFIA, IncFII, p0111	*bla*_NDM-5_, *bla*_CTX-M-15_	100%	n.a.
	W	3 October 2016 °				50%	n.a.
Eco16	H	11 August 2020	1 (92.2)	IncFIA, IncFIB (AP001918), IncFII (pAMA1167-NDM-5)	*bla*_NDM-5_, *bla*_CTX-M-15_	100%	n.a.
	H	16 May 2013				80%	n.a.
	H	9 September 2013				80%	n.a.
	H	16 June 2014				80%	n.a.
	H	22 January 2016				80%	n.a.
	H	8 July 2016				80%	n.a.
	H	27 March 2017				70%	n.a.
	H	21 October 2017				80%	n.a.
	H	30 April 2018				80%	n.a.
	H	3 May 2018				80%	n.a.
	H	11 June 2018 *				80%	n.a.
	H	6 September 2018				80%	n.a.
	H	13 February 2019				70%	n.a.
	H	26 March 2019				75%	n.a.
	H	19 February 2020				70%	n.a.
	W	8 August 2016				100%	n.a.
	W	12 September 2017				100%	n.a.
	W	3 July 2018				90%	n.a.
	W	31 October 2016				70%	n.a.
	W	31 October 2016 *				60%	n.a.

^a^ H = human-retrieved, W = wastewater-retrieved, ^b^ an * indicates exact geographical matches (human cases within the WWTP service area), and a ° indicates near geographical matches (human cases and WWTP maximally 25 km apart as the crow flies); ^c^ no *bla*_CTX-M-15_ detected. Reference strains are indicated by gray shading; n.a. = not applicable.

**Table 3 microorganisms-14-00016-t003:** Detection of plasmids in human-retrieved and wastewater-retrieved CP *K. pneumoniae* clusters.

Cluster	Origin Isolate ^a^	Isolation Date ^b^	Long Read (LR) Contig Characteristics			Results Short Read (SR) Mapping (% Match with)	
			Plasmid	Plasmid Replicon(s)	Carbapenemase and ESBL Genes	Plasmid 1 (Carbapenemase)	Plasmid 2 (CTX-M)
Kpn2	H	20 November 2019 *	1 (114)	IncA/C2, IncFIA(HI1)	*bla* _NDM-1_	100%	
	W	10 October 2017 *				87.4%	
Kpn3	H	28 March 2017 *	1 (335)	pKPC-CAV1321	*bla*_KPC-2_, *bla*_CTX-M-15_	100%	
	W	20 June 2016 *				100%	
Kpn4	H	2 February 2016	1 (164)	IncFIB (pQil), IncFII(K)	*bla* _NDM-1_	100%	n.a.
			2 (38.2)	IncFII, IncR	*bla* _CTX-M-15_	n.a.	100%
	H	24 June 2015				100%	100%
	H	23 November 2015 °				98%	33%
	W	31 October 2016 °				97%	100%
Kpn6	H	22 February 2017	1 (177)	IncA/C2	bla_NDM-1_	100%	n.a.
			2 (174)	IncFIB(K), IncFII(K)	*bla* _CTX-M-15_	n.a.	100%
	H	30 November 2015				100%	100%
	H	1 December 2015				100%	100%
	H	9 December 2015				100%	100%
	H	7 December 2015				100%	100%
	H	7 December 2015				100%	100%
	H	8 December 2015				100%	100%
	H	7 December 2015				100%	100%
	H	17 December 2015				100%	100%
	H	10 December 2015				100%	100%
	H	18 December 2015				100%	100%
	H	18 December 2015				100%	100%
	H	18 December 2015				100%	100%
	H	18 December 2015				100%	100%
	H	18 December 2015				100%	100%
	H	18 December 2015				100%	100%
	H	16 December 2015				100%	100%
	H	9 December 2015				100%	100%
	H	21 December 2015				100%	100%
	H	21 December 2015				100%	100%
	H	18 December 2015				100%	90% ^c^
	H	24 December 2015 °				100%	100%
	H	28 December 2015				100%	89% ^c^
	H	4 January 2016				100%	100%
	H	25 January 2016				100%	93.3%
	H	4 January 2016				96%	100%
	H	8 February 2016				100%	100%
	H	24 February 2016				100%	89%
	H	7 March 2016				100%	91% ^c^
	H	10 March 2016				100%	100%
	H	11 March 2016				99.8%	100%
	H	2 May 2016				100%	100%
	H	19 March 2016				100%	100%
	H	6 October 2016				100%	90% ^c^
	H	15 December 2015				100%	93%
	H	30 October 2018				100%	100%
	W	25 July 2017				100%	100%
	W	18 April 2016 °				100%	100%

^a^ H = human-retrieved, W = wastewater-retrieved, ^b^ an * indicates exact geographical matches (human cases within the WWTP service area), and a ° indicates near geographical matches (human cases and WWTP maximally 25 km apart as the crow flies); ^c^ no *bla*_CTX-M-15_ detected. Reference strains are indicated by gray shading; n.a. = not applicable.

## Data Availability

All sequences have been deposited and are openly available in the NCBI Sequence Read Archive (SRA) database at https://ncbi.nlm.nih.gov/sra (accessed on 7 July 2025). Data on the origin of isolates is listed in Table 1 and Appendix A. More information on the origin of isolates can be retrieved from publications referred to in the article. Further inquiries can be directed to the corresponding author.

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
