# Peer review of "Comparative Genomics of Human- and Wastewater-Derived CPE Isolates in The Netherlands Reveals Shared and Complementary Characteristics"

_microorganisms, 2025, doi:10.3390/microorganisms14010016_

Round 1
Reviewer 1 Report
Comments and Suggestions for Authors
Dear authors,
Your manuscript is overall well written, clearly structured and covers a timely and interesting topic by comparing human and wastewater isolates - in sum, it is a good read. However, I do have some questions/comments, which you will find in the following:
INTRODUCTION
- p. 2, lines 43-49: This is in parts a dublication of text in the methods sections, suggest to drastically shorten it here.
- p. 2, lines 64-66: Sugges to move this part to the method section.
- p. 2, lines 70-72: You explain cleary WHAT you do, but the study could be improved by precisely writing WHY you do that. Suggest to add sentences on the scientific objective of comparing data from these two sources.
- p. 2, line 72: wgMLST, abbreviation not introduced.
MATERIALS AND METHODS
- Overall: comprehensive and clear materials and methods section, but some further information are required.
- p. 3, lines 93: "Unknown locations" due to missing data or how can one understand this?
- p. 3, lines 96-97: Please explain why you used a longer time period fpr the human surveillance, i.e., not restricting it to the same time window as for the WW surveillance.
- p. 3, lines 101-105: Despite being based on other studies, please shortly elaborate on the selection process of the 100 WWTPs, also in regard to geographical distribution and representativeness of the population covered.
- p. 3, lines 101-105: Please provide the number of samples that you considered for your study.
- p. 4, lines 167-169: How well do the ZIP-code areas overlap with the sewersheds of the WWTPs and how may this have affected your spatial analysis, i.e. not considering (un)equal spatial distribution of residential areas within a ZIP-code area?
- p. 5, lines 200-201: Does this belong here?
RESULTS
- p. 8, lines 243: How was this one human CPE isolate selected and has this selection an influence on the results?
- p. 10, lines 275-276: Why is this relevant? Why would one not expect the wgMLST variants to be present regardless of the wastewater studies?
- p. 10, lines 278-279: Would be helpful to add the percentage to the number of clusters (14 out of ? and 9 out of ?)
DISCUSSION
- p. 11, line 299: If this is the main research objective, why use all the data beyond the observation period of the WW surveillance?
- p. 12, lines 361-363: Which roles do other sources of the bacteria play?
Throughout the manuscript: There are a couple of minor language and formatting mistake (e.g. incorrent tense of verbs, double spaces); suggest using a language correction tool.
Reviewer 2 Report
Comments and Suggestions for Authors
Thank you for the opportunity to review this interesting and timely manuscript. The study compares carbapenemase-producing Enterobacterales (CPE) isolated from humans and wastewater in the Netherlands using high-resolution wgMLST and plasmid genomics. This topic is highly relevant to public-health surveillance and contributes meaningfully to ongoing discussions on wastewater-based epidemiology (WBE) for antimicrobial resistance (AMR).
Overall, the manuscript is well written, scientifically sound, and supported by a robust dataset. The work is of high significance, and the conclusions are generally supported by the results. Below are specific comments and suggestions intended to improve clarity, reproducibility, and the overall impact of the manuscript.
Clarify methodological details to improve reproducibility.
Several important aspects of the Methods section require more precision:
Selection criteria for wastewater isolates subjected to whole-genome sequencing.
Sequencing quality metrics (coverage depth, N50, trimming thresholds, quality filtering).
Software versions and specific parameters used for plasmid reconstruction and wgMLST analyses.
Greater methodological transparency would greatly strengthen reproducibility.
Moderate some overinterpretation in the Discussion and Conclusions.
While the data strongly support genomic overlap between human and wastewater isolates, claims suggesting direct individual-level links or shedding events should be softened. The findings are more appropriately interpreted as reflecting population-level circulation rather than direct transmission pathways.
Strengthen discussion of WBE limitations.
The manuscript would benefit from explicitly acknowledging limitations such as:
sparse temporal wastewater sampling,
incomplete geographic resolution of human case metadata,
potential biases in clinical surveillance,
limited ability to infer prevalence or transmission directionality.
Including a short, explicit “Limitations” paragraph would improve balance.
Improve clarity of complex figures.
While the figures are informative, some are visually dense—particularly MSTs and spatiotemporal cluster figures. Enhancing color contrast, enlarging node labels, or providing zoom-in panels for densely connected clusters would help readers interpret the genomic relationships more easily.
English language refinements.
The manuscript is generally well written, but some sentences—especially in the Methods and Discussion—are long or verbose. Minor language polishing would improve readability.
Contextual references.
Consider adding a few additional references addressing global WBE–AMR efforts, sewer microbiology, and previous comparative genomic studies to further strengthen the introduction.
Enhance explanation of clustering thresholds.
The thresholds for wgMLST cluster definitions (≤25 loci for E. coli, ≤20 for K. pneumoniae) should be briefly justified with citations or methodological rationale.
Tables.
Table 2 is rich in content but visually heavy. Consider splitting it by species or adding subtle row shading to improve readability.
This is a high-quality manuscript with a strong dataset and important findings. With minor to moderate revisions—primarily focused on methodological clarity, figure presentation, and cautious interpretation—the paper will be an excellent contribution to the field and suitable for publication.
Comments on the Quality of English LanguageThe manuscript is generally well written and clearly structured, and the scientific content is understandable throughout. However, several sections, particularly within the Methods and Discussion, would benefit from minor language polishing to improve clarity, flow, and conciseness. Some sentences are overly long or contain small grammatical inconsistencies that can be easily corrected. A light professional English editing pass is recommended to ensure maximal clarity and readability.
